# Large Span Sizes and Irregular Shapes Target Detection Methods Using Variable Convolution-Improved YOLOv8

**DOI:** 10.3390/s24082560

**Published:** 2024-04-17

**Authors:** Yan Gao, Wei Liu, Hsiang-Chen Chui, Xiaoming Chen

**Affiliations:** 1School of Intergated Circuits, Dalian University of Technology, Dalian 116024, China; 2642405799@mail.dlut.edu.cn; 2Automation Department, Lingyuan Iron and Steel Group Co., Ltd., Lingyuan 122500, China; liuwei_lgit@sina.com; 3School of Optoelectronic Engineering and Instrumentation Science, Dalian University of Technology, Dalian 116024, China; hcchui@dlut.edu.cn

**Keywords:** small object, classification, improved YOLOv8, steel scrap

## Abstract

In this work, an object detection method using variable convolution-improved YOLOv8 is proposed to solve the problem of low accuracy and low efficiency in detecting spanning and irregularly shaped samples. Aiming at the problems of the irregular shape of a target, the low resolution of labeling frames, dense distribution, and the ease of overlap, a deformable convolution module is added to the original backbone network. This allows the model to deal flexibly with the problem of the insufficient perceptual field of the target corresponding to the detection point, and the situations of leakage and misdetection can be effectively improved. In order to solve the issue that small target detection is susceptible to image background and noise interference, the Sim-AM (simple parameter-free attention mechanism) module is added to the backbone network of YOLOv8, which enhances the attention to the underlying features and, thus, improves the detection accuracy of the model. More importantly, the Sim-AM module does not need to add parameters to the original network, which reduces the computation of the model. To address the problem of complex model structures that can lead to slower detection, the spatial pyramid pooling of the backbone network is replaced with focal modulation networks, which greatly simplifies the computation process. The experimental validation was carried out on the scrap steel dataset containing a large number of targets of multiple shapes and sizes. The results showed that the improved YOLOv8 network model improves the AP (average precision) by 2.1%, the mAP (mean average precision value) by 0.8%, and reduces the FPS (frames per second) by 5.4, which meets the performance requirements of real-time industrial inspection.

## 1. Introduction

In recent years, target detection has become a popular direction in computer vision and digital image processing [1]. However, when recognizing targets that are irregularly shaped and span a wide range of sizes, where smaller and irregularly shaped targets have fewer features, deep learning models require larger datasets to support the computations used for such target detection. In order to improve accuracy, the training set should highly emphasize images that are similar to the actual application scenarios, and even for the same object, different background environments can have a significant impact on the results. How to recognize objects efficiently and accurately with large differences in size and shape is usually a big problem.

The Fire-YOLO detection model expands the feature extraction network from three dimensions, which enhances the feature propagation of small fire target identification, improves network performance, and reduces model parameters [2]. The small-scale feature layer is fused with the second and first feature layers, and the enhanced feature layer is output [3]. Wu, H.Y. et al., based on the YOLOv4 network, modify the residual block of the backbone network into a Res2block residual block with a hierarchical residual mode and construct a new backbone network, Csp2Darknet53, to enhance fine-grained feature detection [4]. The neck network of the YOLOv5 infrared small dark target detection model is developed by fusing the features extracted after one downsampling of the backbone network with the top-level semantic information of the neck network to obtain a target detection head with a small perceptual field [5]. An improved lightweight architecture for real-time small target detection is proposed, based on the You Only Look Once (YOLO) model by combining channel-wise attention block, space-attention block, and multi-branch convolutional neural network structure [6].

Shabbir, S. et al. used a CNN (convolutional neural network) [7] and SEEDS (Super-pixels Extracted via Energy-Driven Sampling) [8] algorithm to achieve the automatic classification of different types of nonferrous metal waste, which provides new possibilities for the automatic classification of metal wastes with high accuracy. Gao, Z.J. et al. employed a CNN to improve the optical recognition of debris obtained from industrial sources and to minimize the effects of surface inhomogeneities, among others [9], which laid the foundation for the classification problem of metal scrap images.

Based on the research literature, none of the existing methods can solve the problem of low accuracy and efficiency in detecting large span sizes and irregular shapes in scrap steel classification. And, most of the current technologies can only distinguish nonferrous metals or plastic sundries in scrap steel but fail to achieve the classification of scrap steel grades. Additionally, the classification of scrap steel into light as shown in Figure 1a, medium as shown in Figure 1b, heavy types as shown in Figure 1c, oily as shown in Figure 1d, and confined as shown in Figure 1e is important because each type has a different price and affects the quality of the final product. Existing target detection algorithms are divided into traditional manual feature extraction detection algorithms and deep learning-based detection algorithms.

The traditional detection algorithm types are the V–J (Viola and Jones) detection algorithm [10], the HOG (Histogram of Oriented Gradients) and SVM (Support Vector Machine) detection algorithm [11], the DPM (Deformable Part-Based Model) algorithm [12], etc. However, there are still many problems with traditional target detection methods because sliding windows bring a large number of redundant windows, which consume a lot of time, and because the robustness and generalization are too poor due to the use of manually extracted features [13], so the traditional target detection algorithms are not suitable for use in industrialized scenarios.

Convolutional neural networks offer a notable advantage over traditional target detection algorithms by automatically extracting features and learning optimal representations, leading to enhanced recognition accuracy. Two primary categories of deep learning-based target detection algorithms exist: two-stage and one-stage. In contrast to the two-stage approach, the one-stage algorithm employs a single network to directly predict object locations and markers. Although the one-stage algorithm exhibits slightly lower detection accuracy, its notable speed advantages make it particularly well-suited for real-time detection scenarios.

Two-stage algorithms mainly include R-CNN series [14], R-FCN [15], Mask R-CNN [16], etc. Zhang W et al. proposed the two-stage target detection algorithm R-CNN [17], but the training speed is difficult to improve, the disk space and time consumed are also relatively large and the testing speed is slow. Zhang Y et al. proposed a spatial pyramid network [18], which led to a significant increase in detection speed. Shim et al. proposed the Fast R-CNN [19] algorithm, which can reach 70% mAP, and the speed has been further improved. The detection accuracy of two-stage target detection is continuously improving, but the speed of detection can only reach about 11 FPS, which still does not meet the requirements of real-time in the industry.

One-stage target detection algorithms, known as regression analysis-based target detection algorithms, are typically represented by the YOLO (You Only Look Once) series and the SSD (Single Shot Multi-Box Detector) series. Joseph Redmon et al. proposed a one-stage target detector YOLO, which can reach a recognition speed of 45 FPS, processing 45 frames per second, and which can easily run in real time and reach an accuracy of detection of 63.4%; however, its recognition accuracy and detection speed cannot meet the detection needs of the industry [20]. The problem of recognizing fast-moving objects was solved using the YOLOv2 algorithm to estimate the optimal fish head position in each frame, and by subsequently connecting the trajectory of each fish between frames [21]. However, this method does not fundamentally solve the problem of real-time detection. Liu X et al. proposed an improved YOLOv3 algorithm, and the experimental results in the COCO dataset showed an increase in mAP of 7.9, but its detection speed and accuracy were still unsatisfactory [22]. Sun, X et al. implemented the real-time inspection of power systems using the YOLOv4 algorithm, but the proposed method is susceptible to the interference of background information, and it is difficult to realize the identification of multiple overlapping targets [23]. Zhang, L. et al. developed the YOLOv5 algorithm to solve the problem of the long-range detection of small target sizes in forest fires, and the accuracy rate was significantly improved, but the method has more parameters, and the detection speed is slower [24]. Pullakandam, M. et al. developed the YOLOv8 algorithm to realize the abnormal behavior detection of surveillance cameras; compared with YOLOv5, the mAP value was increased by 1%, and the detection speed was also greatly improved [25]. Li, Y.T. et al. improved YOLOov8 by introducing the idea of Bi-PAN-FPN, which further solved the problem of small targets in aerial images being easy to misdetect and omit and thereby obtained more satisfactory results [26].

In summary, the YOLOv8 detection algorithm stands out among its counterparts with its notable strengths in high accuracy, speed, and stability, effectively meeting the real-time demands of industrial scenarios. Nevertheless, acknowledged challenges associated with the YOLOv8 algorithm for target detection include its relatively lower accuracy and efficiency when confronted with the recognition of irregularly shaped targets that span a wide range of sizes.

Aiming at overcoming the defects of the above algorithms, this work proposes an improved method of YOLOv8, and the main contributions of this paper are as follows:

(a) Deformable convolutional blocks are added to the backbone network of YOLOv8 so that the model can flexibly deal with the problem of insufficient sensory field for targets of different shapes and sizes, strengthen the focus on small targets, and improve detection accuracy.

(b) The Sim-AM module is integrated into the backbone of YOLOv8, enabling the model to filter out relatively important information from large amounts of data. The problem of the target to be detected being susceptible to image background and noise interference is solved. Importantly, the Sim-AM module introduces no additional parameters, significantly reducing the computational load of the model.

(c) The spatial pyramid pool of the backbone network is replaced by a focal modulation network. This network allows for the separation of aggregates from individual queries, greatly simplifying the computational process of the model and, thus, solving the problem of slow detection due to the complex model structure.

## 2. Methods

In this section, the theoretical foundation of the classical YOLOv8 is expounded upon. Following this foundation, a method for improving the YOLOv8 network model based on variable convolution is proposed.

### 2.1. YOLOv8 Network Model

As can be seen in Figure 2, YOLOv8 comprises a feature extraction network and a multiscale feature fusion network. The computational process begins with a feature extraction network to analyze the input image and extract relevant features. Subsequently, target detection is performed with a multiscale feature fusion network, which enables the model to capture information at different scales. Finally, the output consists of four parameters representing the coordinates of the bounding box, as well as the confidence and category probabilities of the predicted frames.

#### 2.1.1. Data Preprocessing

The data preprocessing of YOLOv8 still adopts the strategy of YOLOv5, using four enhancement methods including Mosaic, Mixup, random perspective, and HSV augment [27].

#### 2.1.2. YOLOv8 Backbone Network

The backbone network structure of YOLOv8 is reminiscent of YOLOv5, which has a distinct architectural pattern. In general, it consists of downsampling the feature map using a 3 × 3 convolutional layer with a step size of 2, followed by further enhancement of the features using C3 modules. In YOLOv8, this general model is largely retained, but the original C3 module has been replaced by the new C2f module. The C2f module in YOLOv8 introduces additional branches to enrich the gradient flow during backpropagation. The following figure illustrates the C2f module in YOLOv8 and the C3 module in YOLOv5, as shown in Figure 3.

YOLOv8 adheres to the traditional FPN + PAN structure for constructing a feature pyramid network, facilitating the comprehensive integration of multiscale information. While the C3 module in FPN-PAN is replaced by the C2f module, the remaining components closely resemble the FPN-PAN structure in YOLOv5.

From YOLOv3 to YOLOv5, the detection heads were coupled, meaning a single convolutional layer was employed to simultaneously handle both classification and localization tasks. It was not until the advent of YOLOX that the YOLO family embraced the decoupled head structure. YOLOv8 also adopts the decoupled head structure, where two parallel branches independently extract category and location features. Subsequently, each branch utilizes a 1 × 1 convolutional layer to accomplish the classification and localization tasks.

#### 2.1.3. Label Assignment Strategy

In terms of label assignment strategy, although YOLOv5 incorporates some functionality for automatically clustering candidate boxes, the effectiveness of these clustered boxes relies on the dataset. If the dataset is not sufficiently diverse and fails to accurately reflect the distribution characteristics of the data, the clustered candidate boxes may exhibit significant disparities in size proportions compared to real-world object dimensions. YOLOv8 does not employ a candidate box strategy, and, thus, the problem addressed is related to the multiscale assignment of positive and negative samples. Unlike SimOTA used in YOLOX, YOLOv8 adopts the same task ownership and object detection strategy as YOLOv6 for label assignment, which is a dynamic label assignment approach. YOLOv8 utilizes only the ${target}_{bboxes}$ and ${target}_{scores}$, without considering the prediction of whether an object is present. The losses in YOLOv8 primarily consist of classification loss and position loss. For YOLOv8, the classification loss is formulated as Varifocal Loss (VFL), while the regression loss takes the form of Complete Intersection over Union Loss (CIoU) and Distributional Feature Loss (DFL). The Varifocal Loss is defined as Equation (1).
(1)$$VFL(p,q)=\left\{\begin{array}{cc} -q(q\log(p)+(1-q)\log(1-q)) & q>0\\ -\alpha p^{\gamma }\log(1-p) & q=0 \end{array}\right.$$
where p represents the predictive classification score, q∈[0,1] and q represents the predicted target score. If it corresponds to the true class, q is the Intersection over Union (IoU) between the prediction and the ground truth. If it pertains to another class, the value of q is set to 0. The VFL Loss employs asymmetric parameters to weigh positive and negative samples differently. It achieves this by selectively attenuating only the negative samples, thereby introducing asymmetry in how the foreground and the background contribute to the loss. For positive samples, weighting is applied based on the parameter q. If the Ground-Truth IoU (${GT}_{IoU}$) of a positive sample is high, its contribution to the loss is greater. This allows the network to focus more on high-quality samples during training, emphasizing that training on high-quality positive samples has a more substantial impact on improving average precision (AP) compared to lower-quality positives. For negative samples, a weighting factor of $p^{\gamma }$ is employed to reduce their contribution. This reduction is effective because the predicted score p for negative samples tends to decrease after taking the power γ. Consequently, this diminishes the overall contribution of negative samples to the loss.

### 2.2. YOLOv8 Network Model Improved Based on Adaptive Convolution

#### 2.2.1. Deformable Convolution Network

A Deformable Convolution Network (DCN) [28] series of algorithms renders the convolution kernel to no longer be a simple rectangle but learn an offset at each point, so that the convolution kernel can learn different convolution kernel structures according to different data, thus enhancing the ability of the model to learn the invariance of the complex target, as shown in Figure 4.

The traditional convolutional computation process involves sampling a set of pixel points from the input feature map and then using a convolutional operation to compute the sampled results; the result can be obtained as Equation (2).
(2)$$y(p_{0})=\sum_{p_{n}\in R}w(p_{n})\cdot x(p_{0}+p_{n})$$

The deformable convolution, on the other hand, calculates the offset of the result through the sampling method of bilinear interpolation to achieve the effect of deforming the convolution kernel. $\Delta p_{n}$ denotes the offset of $p_{n}$ as Equation (3).
(3)$$y(p_{0})=\sum_{p_{n}\in R}w(p_{n})\cdot x(p_{0}+p_{n}+\Delta p_{n})$$

#### 2.2.2. Simple Parameter-Free Attention Mechanism

Conventional modules of attentional mechanisms are mainly categorized into spatial and channel domains, but in computer vision, these two mechanisms should jointly participate in information selection during visual processing. It is challenging to estimate the complete 3D weights directly. The Convolutional Block Attention Module (CBAM) estimates the 1D and 2D weights separately after combining them, which consumes a large amount of computational time. The advantage of the Sim-AM [29] module is that it directly infers the 3D weights from the current neurons, and then goes back to optimize these neurons, which improves the simultaneous change in the spatial and channel domains with increased flexibility and reduces the computational load of the weights. In other words, neurons that exhibit significant spatial inhibition effects should be given higher priority in visual processing as Equation (4).
(4)$$e_{t}(w_{t},b_{t},y,x_{i})={\left(y_{t}-\overset{\Lambda }{t}\right)}^{2}+\frac{1}{M-1}{\sum_{i=1}^{M-1}\left(y_{0}-\overset{\Lambda }{x_{i}}\right)}^{2}$$
where =wtt+tΛbt and =wtxi+xiΛbt are linear transformations of t and $x_{i}$; t and $x_{i}$ are the target neuron and other neurons in a single channel with input feature $X\epsilon R^{C*H*W}$; i is an exponent in the spatial dimension; M=H∗W is the number of neurons in that channel; and $w_{t}$ and $b_{t}$ are transformations of the weights and the bias. The lower the value of $e_{t}^{*}$, the more different the neuron is from the surrounding neurons and the more important for visual processing. Consequently, the importance of each neuron can be obtained by $1/e_{t}^{*}$.

#### 2.2.3. Focal Modulation Networks

Focal modulation uses a set of depthwise convolutional implementations to encode short-range to long-range visual contexts at different levels of granularity, selectively aggregates the contextual features of each token according to its content, and fuses the aggregated features into the query, which greatly simplifies the computation process compared to the traditional self-attention module [30]. Focal modulation generates refined representation $y_{i}$ using an early aggregation procedure as Equation (5).
(5)$$y_{i}=q(x_{i})⊙h(\sum_{l=1}^{L+1}g_{i}^{l}\cdot z_{i}^{l})$$
where $g_{i}^{l}$ and $z_{i}^{l}$ are the gating values and visual features, and $q(x_{i})$ is a query projection function.

#### 2.2.4. Large Span Sizes and Irregular Shapes Target Detection Methods

Based on the above analysis, the backbone network of the traditional YOLOv8 algorithm is improved to cope with complex scrap classification scenarios, and the structure is shown in Figure 5 with the red part being the improved part.

The C2f module is replaced by C2f_DCN in the backbone network of YOLOv8. Since the variable convolution can adaptively adjust the object’s receptive field, it effectively solves the problem of the receptive field size of the scrap steel dataset being different when the scrap steel dataset is in different positional regions of a map. When sampling, C2f_DCN is closer to the size and shape of the object and is more robust, whereas ordinary convolution cannot realize it. Furthermore, the oil-contaminated type of scrap steel targets is small in size and irregular in shape, which may lead to some small targets not being detected if the traditional convolution is still adopted, thus affecting the performance of the model.

The Sim-AM attention mechanism module is added to the end of the YOLOv8 backbone network to adaptively select and adjust the channel weights and spatial weights of the feature maps to better capture and represent important features in the image. More importantly, compared with the traditional attention mechanism model, Sim-AM does not introduce additional parameters for computation, making the model more lightweight and improving computational efficiency.

The SPPF module is replaced by the focal modulation module, which encodes spatial contexts at different granularity levels and then selectively aggregates them according to the query. It greatly simplifies the computational process and makes the model more lightweight, in order to meet the demand for real-time scrap classification and detection in the industry.

## 3. Experiment and Result

### 3.1. Dataset Creation and Preprocessing

Considering the practical application scenarios of steel scrap classification, the data used in this study were obtained from the Ling-yuan Iron and Steel Group, with a total of 16,071 photos and 3 video recordings of real-time scrap steel unloading in industrial scenarios, where scrap is classified into five types, including light scrap, medium scrap, heavy scrap, oily, and confined. In order to complete the training and validation of the YOLOv8 model, 20% of the datasets from all datasets are randomly generated as the test dataset as shown in Figure 6, where Figure 6a is a multi-target data sample and Figure 6b is a single-target data sample.

The data preprocessing part of the training mainly used hybrid enhancement technology, mosaic enhancement technology, spatial perturbation technology, color perturbation technology, and other methods for training data preprocessing. After preprocessing, not only can the training dataset be effectively expanded and the diversity of data increased but also the overfitting problem of the model can be reduced and the generalization ability of the model improved; the preprocessed data samples are shown in Figure 7.

The preprocessed image data of scrap steel are the training data of this experiment. This experiment uses the Pytorch framework, and the accelerated training was carried out using GPU; the specific training parameters of the network model are shown in Table 1.

### 3.2. Evaluation Indicators

The evaluation criteria for target detection contain precision, recall, average precision, and average precision mean. P is the accuracy, expressed as the ratio of the number of scraps correctly predicted for a certain type of scrap to the overall number predicted for that type of scrap, as in Equation (6).
(6)$$P=\frac{TP}{TP+FP}$$

R represents the recall rate, which represents the proportion of the number of scraps correctly detected as a certain type of scrap to the number of all scraps of that type in the test set, as in Equation (7).
(7)$$R=\frac{TP}{TP+FN}$$

The specific concepts of TP, FP, FN, and TN in the precision and recall formulas are as follows, and the correspondence can be seen in Table 2. TP indicates the number of samples that are positive cases and are correctly classified as positive cases. FP indicates the number of samples that are negative cases but are actually classified as positive cases. FN indicates the number of samples that are positive cases and are classified as negative cases alone. TN indicates the number of samples that are negative cases and are correctly classified as negative cases.

AP is the average accuracy, calculated by plotting the PR curve and then selecting the average of the maximum accuracy greater than or equal to each R value. mAP is the average of the AP for each type of scrap, as in Equation (8).
(8)$$mAP=\frac{\sum AP}{N}$$

### 3.3. Convergence Analysis

In order to verify the convergence performance of the optimized model on the dataset, the convergence of the YOLOv8 model and the improved YOLOv8 model is compared. Even the bounding box loss for both models is used to calculate the difference between the predicted bounding box and the true bounding box; the confidence loss is used to calculate the difference between predicted and true feature points and the classification loss is used to compare the difference between predicted category distributions. The true category labels are shown in Figure 8, Figure 9 and Figure 10.

By observing the loss curves of the two models, it can be found that the improved YOLOv8 model converges slower than the YOLOv8 model when the epoch is less than 600 times, and the loss value is higher than the YOLOVv8 model. The improved YOLOv8 model converges faster than the YOLOv8 model when the epoch is greater than 600 rounds, and the loss value is slightly lower than the YOLOV8 model when the epoch reaches 1000 times.

The above results effectively demonstrate that the DCN can effectively reduce the error of bounding box loss, making the localization and generation of bounding box more accurate. The Sim-AM attention mechanism, on the other hand, enhances the degree of attention to the underlying features, which has a more positive impact on the confidence loss, resulting in a smaller discrepancy between predicted and real feature points. The lower classification loss of the improved network model also further illustrates the effectiveness of deformable convolution and Sim-AM for the scrap steel classification task and also represents a better robustness of the model.

### 3.4. Comparative Analysis of Detection Performance

The detection performance of the traditional YOLOv8 network model and the improved YOLOv8 network model were compared in the experiments, and the precision–recall performance curves are shown in Figure 11 and Figure 12. The larger the area enclosed by this curve with the axes, the higher the AP value is, and the higher the mAP value is, the better the performance of the model. The mAP value of the improved model and the AP values of the five types of wastes are improved, the mAP value is improved by 0.8%, the accuracy of light metal wastes is improved by 1%, the accuracy of medium metal wastes is improved by 1.2%, the accuracy of heavy metal wastes is improved by 0.4%, the confined metal wastes are improved by 1.5%, and the oily metal wastes are improved by 0.1%.

In the scrap dataset used in this experiment, light scrap, medium scrap, and confined scrap contain a large number of targets with different sizes and shapes. Based on the above experimental results, it can be seen that the method of deformable convolution proposed in this work can effectively deal with this type of target, as mAP metrics have been significantly improved. On the other hand, the mAP of oil scrap steel is also improved, which further proves the effectiveness of the Sim-AM for dealing with smaller targets and those susceptible to background interference.

The comparisons of the precision curves of the classical YOLOv8 network model and the improved YOLOv8 network model on five types of scrap categories are shown in Figure 13 and Figure 14. According to the experimental results, compared with the classical YOLOv8 model in terms of precision, the precision of each type of scrap is improved in the improved network model, and the average precision is increased by 2.1%. It is worth noting that the precision curve of the improved model is smoother, which is most obvious in the case of oil scrap. This proves that the improved network model has not only improved in accuracy but also has increased robustness.

The comparisons of the recall curves of the classical YOLOv8 network model and the improved YOLOv8 network model on the five types of waste categories are shown in Figure 15 and Figure 16. The recall of the improved YOLOv8 model is improved by 1% compared to the classical model. Based on the above results, the improved YOLOv8 network model, also for light scrap, medium scrap, and closed object scrap, has a more significant increase in the recall rate and a smoother curve. It is further demonstrated that the improved model greatly reduces the miss detection probability of the model for similar targets of different shapes and sizes. However, it is worth noting that due to the small number of samples of oil scrap during training and the interference of extremely similar data samples such as rust and shadows, the recall of oil scrap is still the lowest among the five types, although Sim-AM has slightly improved the recall of the oil-type scrap.

The experimental results of the two-stage detection algorithms SSD; Faster-RCNN; the classical YOLO, YOLOv3, YOLOv5, and YOLOv8 detection algorithms; and the improved YOLOv8 detection algorithm on the scrap steel dataset are compared and analyzed as shown in Table 3. According to the experimental results, the two-stage detection algorithm has a better performance than the classic YOLOv5 algorithm, but this type of detection algorithm has a slower detection speed, which cannot meet the industrial demand for real-time detection. With the continuous iterative development of YOLO series algorithms, newer algorithms tend to have better detection performance. The improved YOLOv8 algorithm has the highest mAP of 93.0%, which is 10.8% higher than the mAP of the classical YOLOv5 network model; the improved YOLOv8 algorithm likewise has a 14% higher recall, and 2.2% higher accuracy that YOLOv5. These rates are also 0.8% higher than the mAP of the classical YOLOv8 network model, 1% higher than its recall, and 2.1% higher than its accuracy. It is worth noting that the introduction of the focal modulation network greatly improves the detection speed of the model, which is 0.003 s per image. A comprehensive comparison of the experimental results can conclude that the improved YOLOv8 network model has the best effect, fast detection speed, and can meet the needs of real-time classification and detection for industrial steel scrap.

### 3.5. Generalization Experiment

In order to further validate the stability of the improved model proposed in this work, a comparative analysis of the performance of YOLOv8 and the improved YOLOv8 model on the publicly available PASCAL VOC dataset was performed; precision–recall performance curves are shown in Figure 17 and Figure 18.

The experimental results show that the performance of the improved YOLOv8 network model has an extremely significant improvement, with a 15.5% increase in mAP on the public dataset compared to the classical YOLOv8 model. It is worth noting that the improved model has a more significant advantage in dealing with samples of different shapes and sizes, such as sofas, where the mAP of the sofa sample is improved by 27.9%. Such results further demonstrate the effectiveness of the proposed method and the stability of the model.

### 3.6. Detection Effect

The experiment comparing the detection effect of the algorithms using the scrap steel dataset is shown in Figure 19. The left side shows the actual labeling results and the right side shows the predicted results of the model.

Through comparative analysis, the improved YOLOv8 detection model can accurately identify scrap steel with different shapes and sizes, of which the more typical is the green-labeled oil-contaminated type of scrap steel in Figure 19, compared with the actual label on the left, which does not occur in the case of missed detection and misdetection. It can be proved that the proposed method is effective in detecting scrap steel datasets with large span sizes and irregular shapes.

## 4. Conclusions

In this work, in order to cope with the low resolution of small target, dense distribution, susceptibility of the image background and noise, deformable convolution and Sim-AM modules are integrated into the backbone network of YOLOv8. This improvement enables the model to adaptively and dynamically handle similar objects of different shapes and sizes, thus improving the detection accuracy of the model. In addition, to address the slow detection speed attributed to a complex model structure, the spatial pyramid pooling in the backbone network has been replaced with a focal modulation network, which improves the speed and efficiency of model detection.

Experimental results based on the scrap dataset show that the improved YOLOv8 network model results in a 2.1% increase in AP, a 0.8% increase in mAP, and a 5.4 reduction in FPS. It verifies that the variable convolution-improved YOLOv8 network model proposed in this work has a better performance in large span sizes and irregular shape target recognition, and it can be widely applied to a similar target recognition problem.

Through the experimental analysis in this work, the recognition accuracy of oil scrap can be further improved in complex scrap datasets. The oil scrap is easily interfered through environmental factors such as rust and shadows, and the small percentage of this type of data samples in the dataset leads to difficulties in classification and detection. In order to solve this problem, a classifier fusion approach can be considered in the future, where the oil scrap is first recognized using a specific classifier, and then fused with the classifiers of easy-to-classify samples.

## Figures and Tables

**Figure 1 sensors-24-02560-f001:**
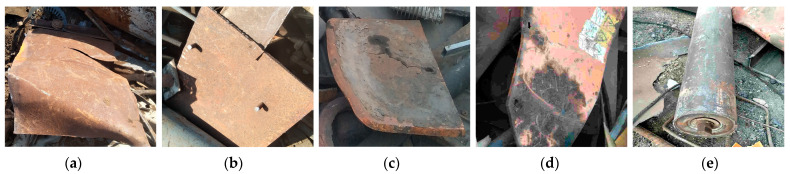
The wheel tread defect types. (**a**) Light scrap steel; (**b**) medium scrap steel; (**c**) heavy scrap steel; (**d**) oily scrap steel; and (**e**) confined scrap steel.

**Figure 2 sensors-24-02560-f002:**
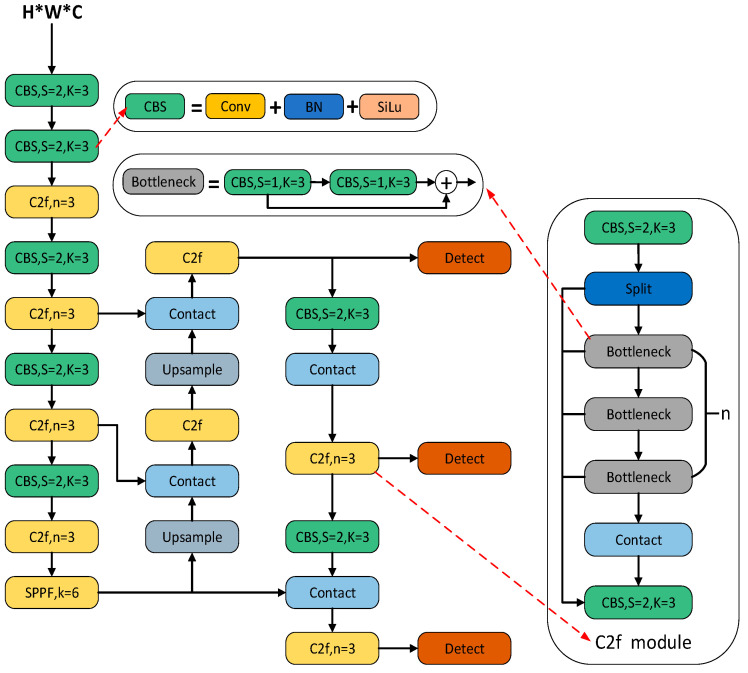
Network structure of YOLOV8 algorithm.

**Figure 3 sensors-24-02560-f003:**
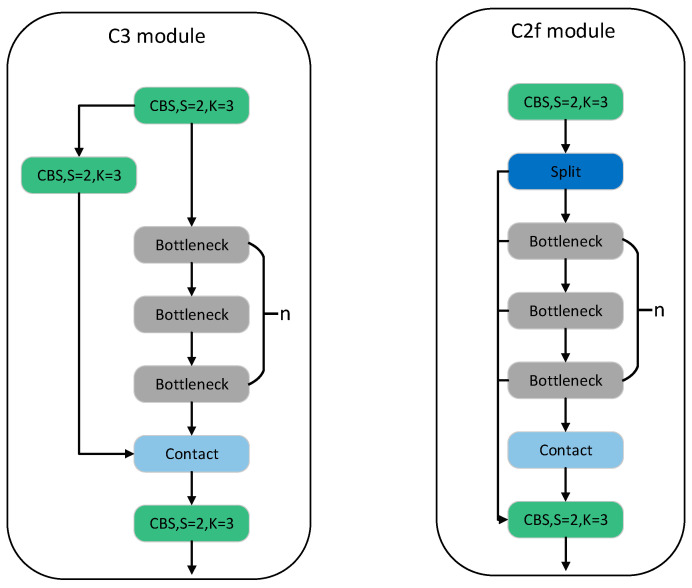
C2f Module and C3 Module.

**Figure 4 sensors-24-02560-f004:**
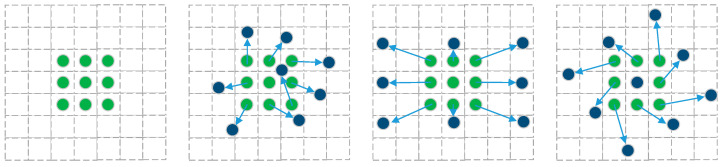
Deformable Convolution Model.

**Figure 5 sensors-24-02560-f005:**
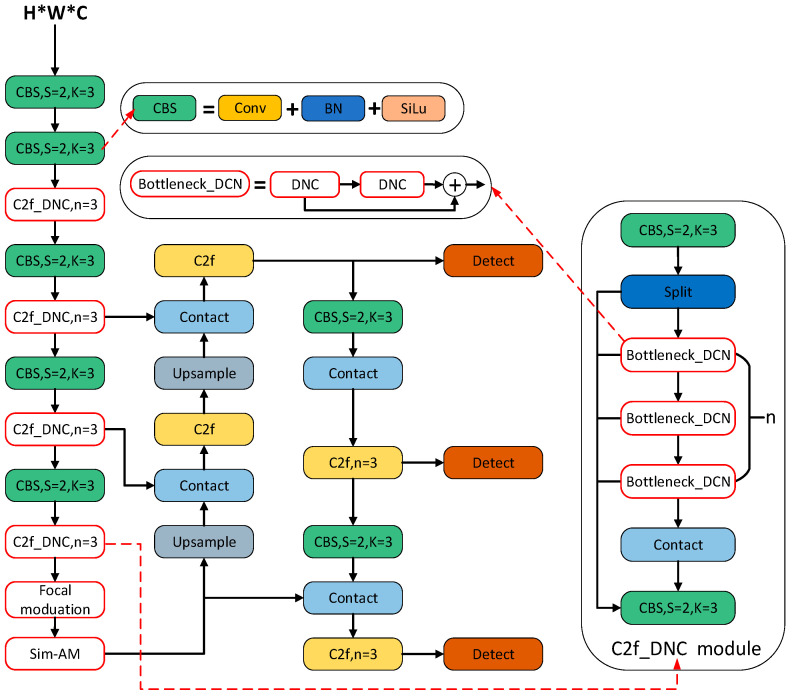
Improved YOLOv8 algorithm network structure.

**Figure 6 sensors-24-02560-f006:**
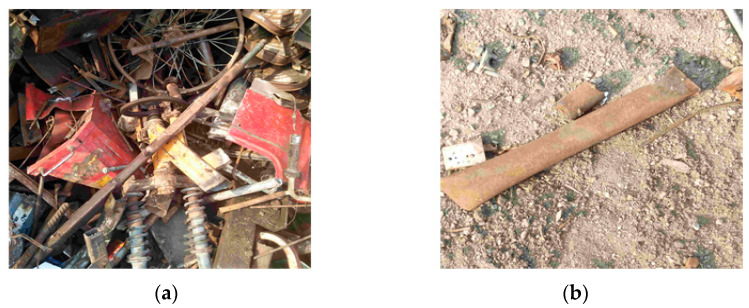
Scrap steel datasets for industrial scenarios. (**a**) A data set containing multiple targets. (**b**) A data set containing a single target.

**Figure 7 sensors-24-02560-f007:**
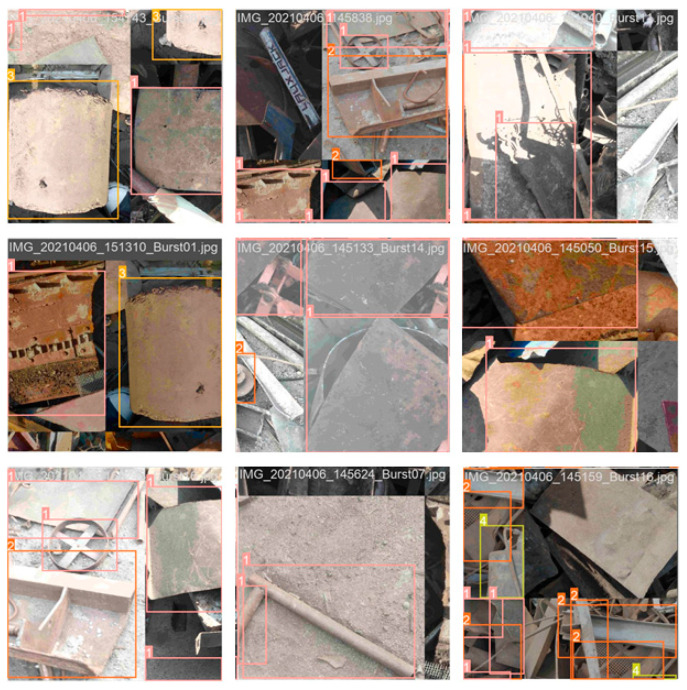
Sample data preprocessing.

**Figure 8 sensors-24-02560-f008:**
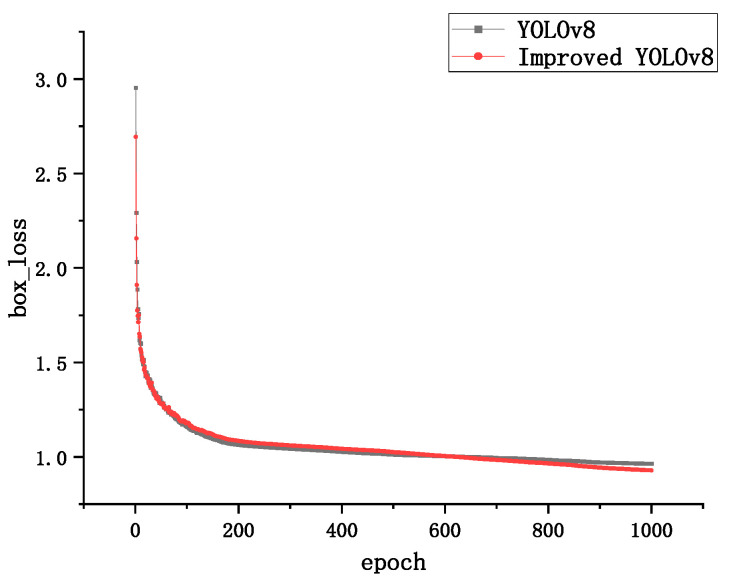
Boundary frame loss curves.

**Figure 9 sensors-24-02560-f009:**
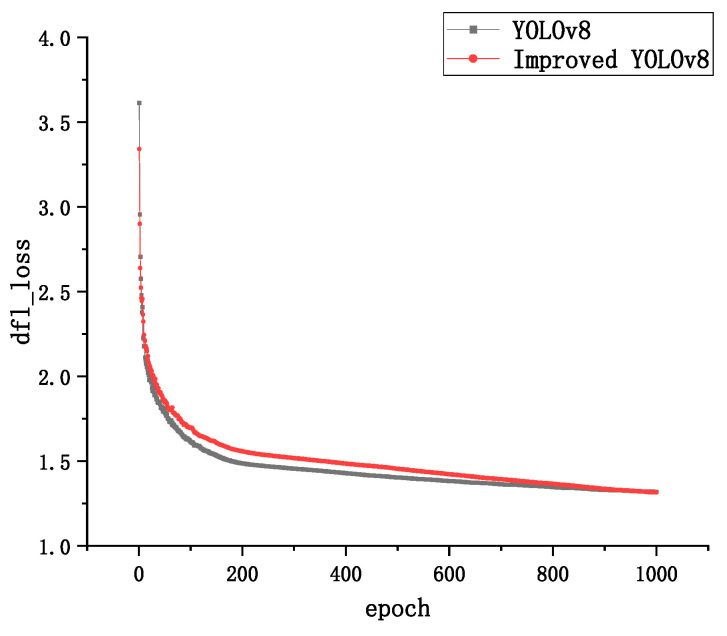
Confidence loss curve.

**Figure 10 sensors-24-02560-f010:**
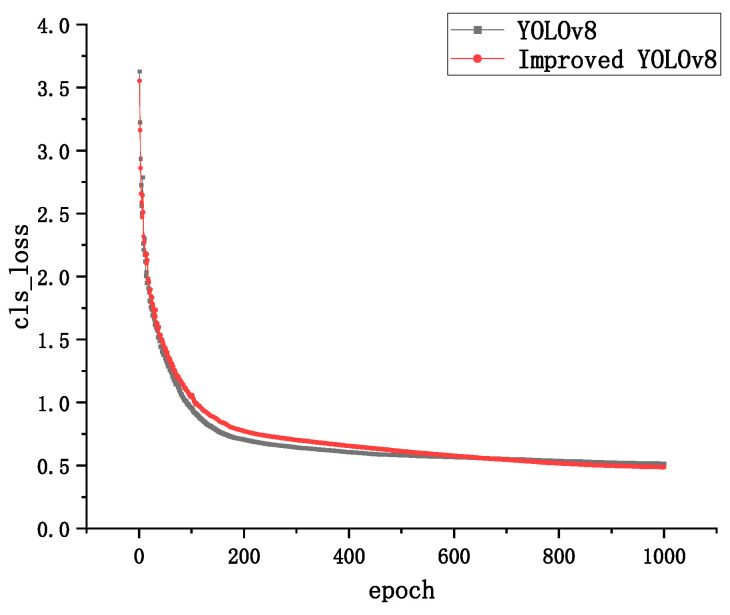
Classification accuracy assessment.

**Figure 11 sensors-24-02560-f011:**
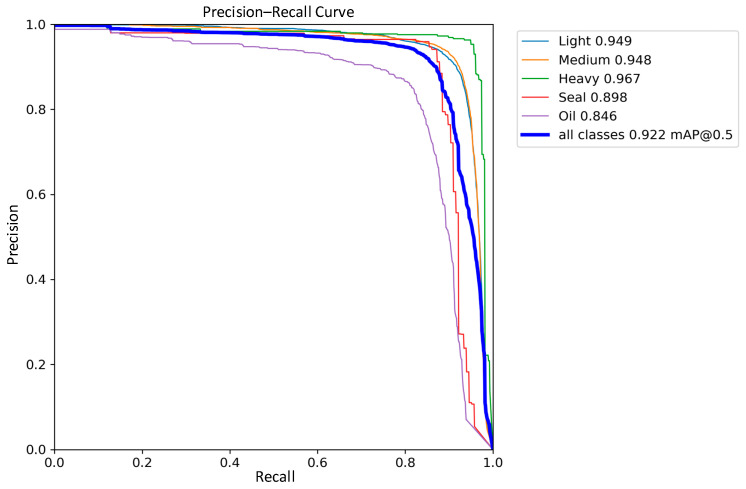
YOLOv8 Precision–Recall Curve.

**Figure 12 sensors-24-02560-f012:**
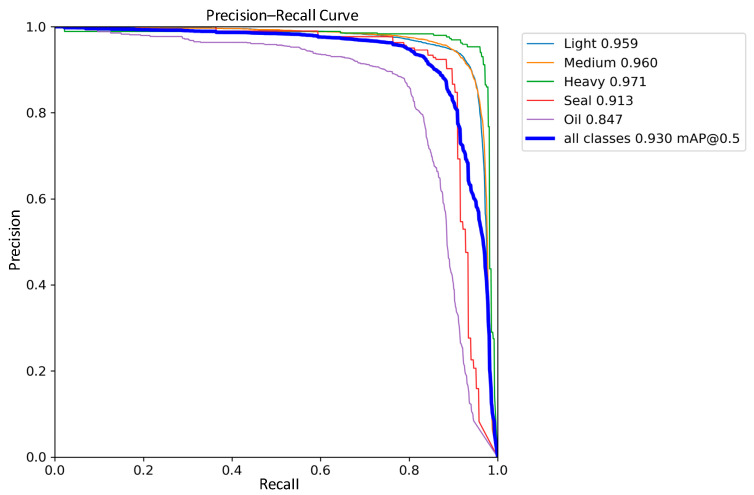
Improved YOLOv8 Precision–Recall Curve.

**Figure 13 sensors-24-02560-f013:**
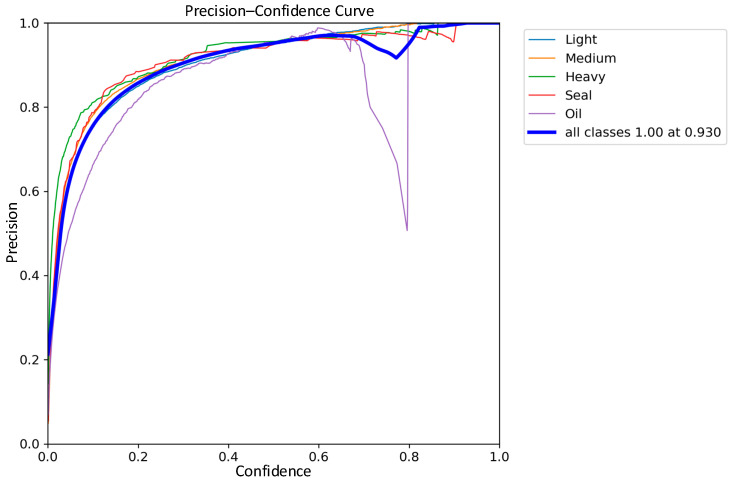
YOLOv8 Precision Curve.

**Figure 14 sensors-24-02560-f014:**
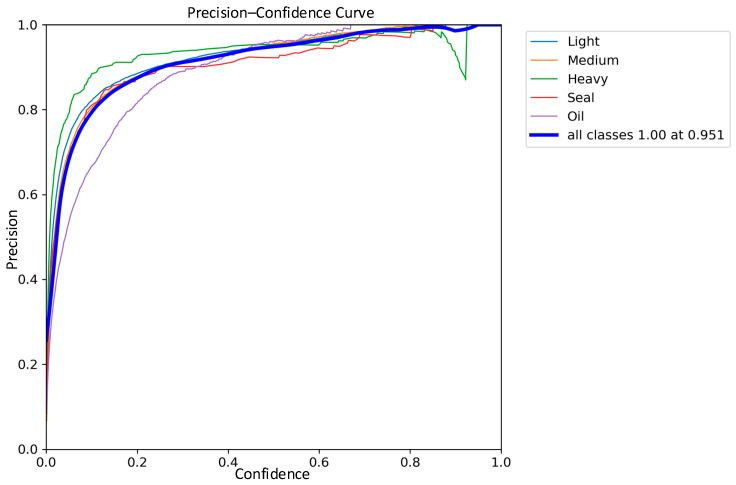
Improved YOLOv8 Precision Curve.

**Figure 15 sensors-24-02560-f015:**
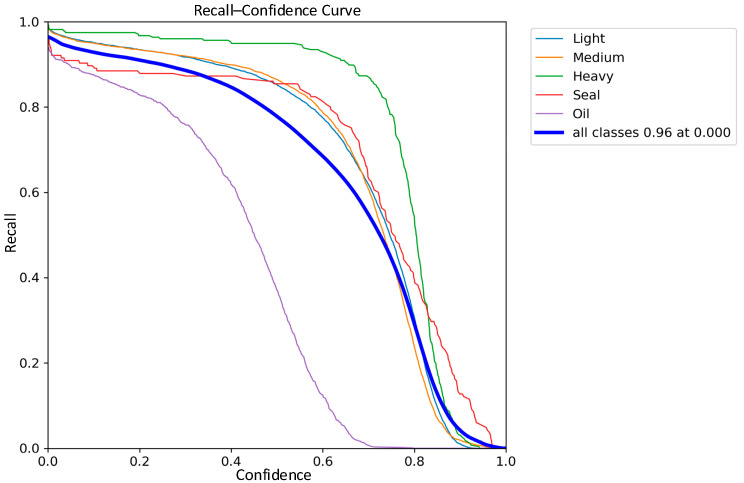
YOLOv8 Recall–Confidence Curve.

**Figure 16 sensors-24-02560-f016:**
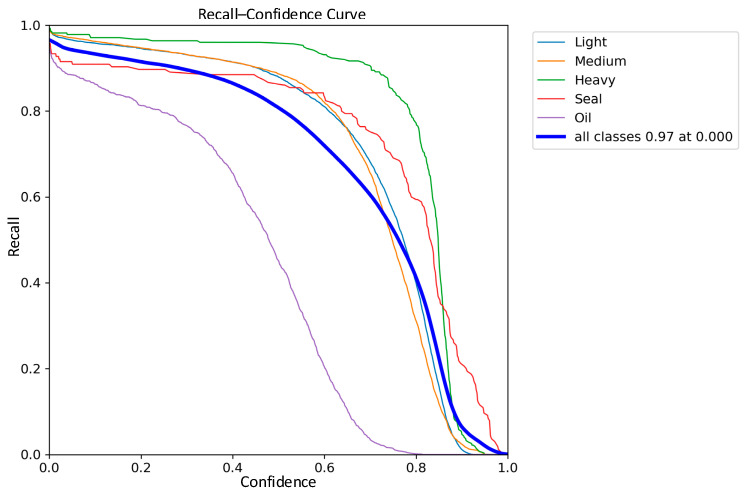
Improved YOLOv8 Recall–Confidence Curve.

**Figure 17 sensors-24-02560-f017:**
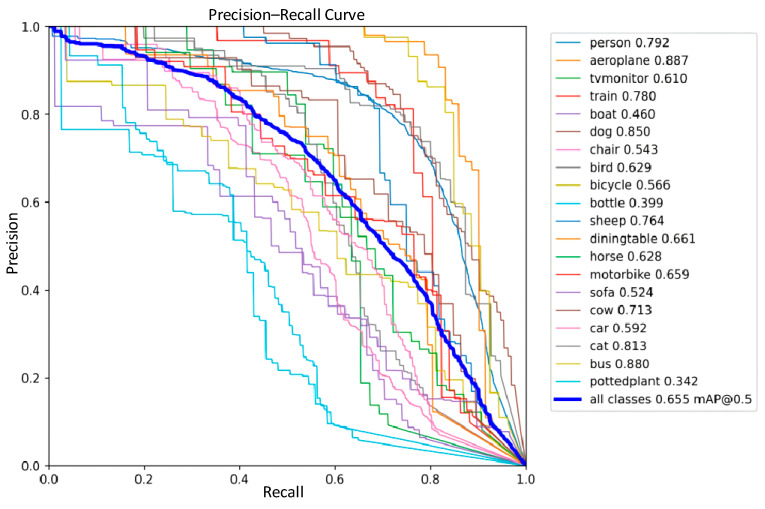
YOLOv8 Precision–Recall Curve in the Publicly Available Dataset.

**Figure 18 sensors-24-02560-f018:**
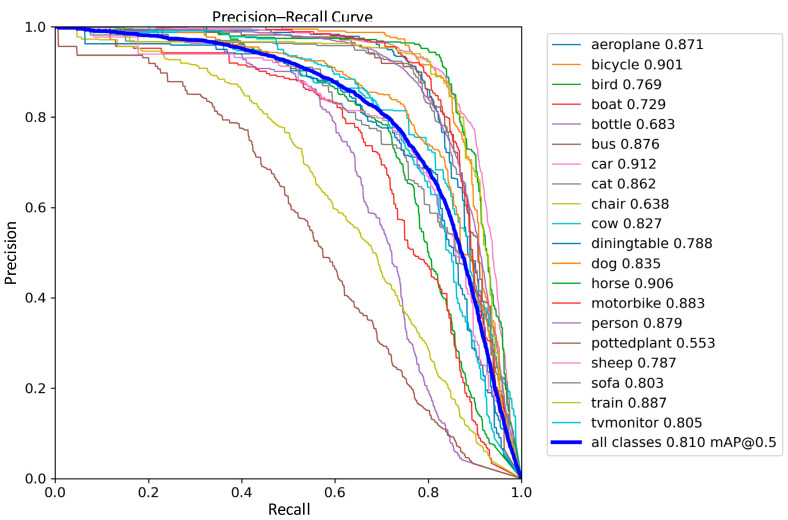
Improved YOLOv8 Precision–Recall Curve in the Publicly Available Dataset.

**Figure 19 sensors-24-02560-f019:**
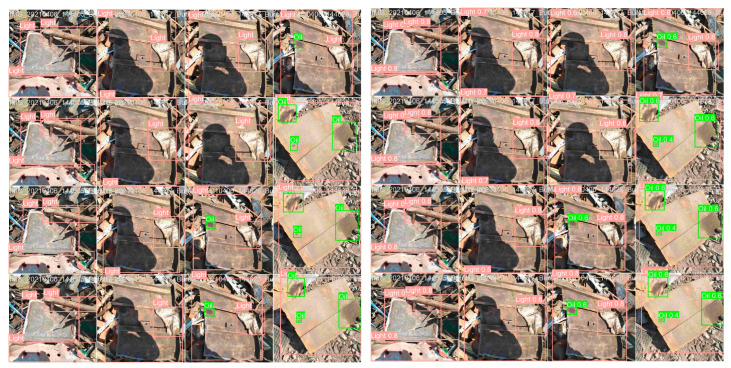
Detection of improved YOLOV8 network model.

**Table 1 sensors-24-02560-t001:** Network model training parameters.

Parameters	Value
Input image resolution	640 × 640 × 3
Epoch	1000
Learning rate	0.01
Batch size	8

**Table 2 sensors-24-02560-t002:** Table of positive and negative cases.

	P (Positive)	N (Negative)
T (True)	True positive (TP)	True negative (TN)
F (False)	False positive (FP)	False negative (FN)

**Table 3 sensors-24-02560-t003:** The comparative analysis of the developed method and other methods.

Framework	mAP	Recall Rate	Accuracy
SSD	84.9%	77.8%	90.1%
Faster-RCNN	90.1%	86.9%	91.2%
YOLO	78.3%	71.2%	87.5%
YOLOv3	80.7%	74.7%	89.2%
YOLOv5	82.2%	83.0%	92.9%
Classic YOLOv8	92.2%	96.0%	93.0%
Improved YOLOv8	93.0%	97.0%	95.1%

## Data Availability

Data are contained within the article.

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
