# Peer review of "Large Span Sizes and Irregular Shapes Target Detection Methods Using Variable Convolution-Improved YOLOv8"

_sensors, 2024, doi:10.3390/s24082560_

Round 1
Reviewer 1 Report
Comments and Suggestions for Authors
1. Only the model experiment results of YOLOV5 are compared in Table 2, and it is suggested to supplement the rich experiment results of other YOLO.
2. The content of the experiment is too little. The author only compares the improved YOLOV8 with the classic YOLOV8. It is necessary to supplement the ablation experiment to show that the improvement of each part has a positive impact on the experimental results.
3, authors are advised to beautify flowcharts
4. It is suggested to supplement the generalization experiment on the public data set to prove the stability of the model.
5. It is suggested that the author supplement the shortcomings in this field in the conclusion and explore the next research direction.
Comments on the Quality of English Languageno
Author Response
Dear Editors and Reviewers:
Thank you for your letter and for the reviewer’s comments concerning our manuscript entitled “Large span sizes and irregular shapes target detection methods using variable convolution improved YOLOv8”. Those comments are all valuable and extremely helpful for revising and improving our paper. We have read comments carefully and have made correction which we hope to meet with approval. The main corrections in the paper and the responds to the reviewer's comments are as flowing:
Comment 1. Only the model experiment results of YOLOv5 are compared in Table 2, and it is suggested to supplement the rich experiment results of other YOLO.
The author’s answer:
In Table 3 (with the addition of model parameter Table 1 and the modification of the original Table 2 to Table 3), the experimental results of the SSD, Faster-RCNN, YOLO, YOLOv3, YOLOv5, YOLOv8, and improved YOLOv8 models on the scrap steel dataset are compared and analyzed.
Comment 2. The content of the experiment is too little. The author only compares the improved YOLOv8 with the classic YOLOv8. It is necessary to supplement the ablation experiment to show that the improvement of each part has a positive impact on the experimental results.
The author’s answer:
Through the comparison and analysis in Table 3, the classical YOLOv8 network model has a significant advantage over other network models, so only the classical YOLOv8 and the improved YOLOv8 are compared and analyzed in more detail. And the positive influence and practical significance of the improvement of each part on the experimental results are elaborated in detail in the revised manuscript.
Comment 3. authors are advised to beautify flowcharts.
The author’s answer:
All flowcharts included in the manuscript have been optimized.
Comment 4. It is suggested to supplement the generalization experiment on the public data set to prove the stability of the model.
The author’s answer:
Subsection 3.6 was added to the revised manuscript to demonstrate the stability of the model by performing generalization experiments on a public dataset.
Comment 5. It is suggested that the author supplement the shortcomings in this field in the conclusion and explore the next research direction.
The author’s answer:
The presentation of the conclusions has been modified in the revised manuscript's, and shortcomings in the field, and next research direction have been elaborated.
We believe that the changes we have made address the concerns raised and improve the quality of the manuscript. Please review the revised version and all major changes have been highlighted in the manuscript. We are grateful for your thorough review and look forward to your feedback.
Thank you once again for your time and expertise.
Yours sincerely

Reviewer 2 Report
Comments and Suggestions for Authors
The manuscript provides an explanation of the proposed enhancements to the YOLOv8 object detection technique, but there are areas that could be improved.
When conducting research, it is important to clearly outline factors such as dataset selection, data preprocessing, model training parameters, and evaluation metrics to ensure that results can be reproduced and are valid. While it is noted that there have been improvements in performance metrics, the actual significance of these improvements is not fully explained. To enhance the credibility of the claims being made, it would be beneficial to include statistical analyses or comparisons with current methods.
The proposed method has its strengths, but there is little talk about its limitations or possible hurdles. Recognizing these factors would offer a more even-handed evaluation of the approach. The language can be a bit wordy and repetitive in spots. Making the wording simpler and keeping the terminology consistent would make it easier to read and understand. The conclusion could be improved by summarizing the main findings and the impact of the study more directly. It would also be helpful to suggest future research directions to give the work a sense of closure and context.
Author Response
Dear Editors and Reviewers:
Thank you for your letter and for the reviewer’s comments concerning our manuscript entitled “Large span sizes and irregular shapes target detection methods using variable convolution improved YOLOv8”. Those comments are all valuable and extremely helpful for revising and improving our paper. We have read comments carefully and have made correction which we hope to meet with approval. The main corrections in the paper and the responds to the reviewer's comments are as flowing:
Comment 1. The manuscript provides an explanation of the proposed enhancements to the YOLOv8 object detection technique, but there are areas that could be improved. When conducting research, it is important to clearly outline factors such as dataset selection data preprocessing, model training parameters, and evaluation metrics to ensure that results can be reproduced and are valid. While it is noted that there have been improvements in performance metrics, the actual significance of these improvements is not fully explained. To enhance the credibility of the claims being made, it would be beneficial to include statistical analyses or comparisons with current methods.
The author’s answer:
In the section on dataset elaboration, the creation of the dataset, the data preprocessing, and the training parameters of the model are elaborated more completely to ensure that the results are reproducible and valid. In the revised manuscript, the positive impact and practical implications of the improvements in each section on the experimental results are detailed. And in Table 3 the improved method, compared and analyzed with current methods.
Comment 2. The proposed method has its strengths, but there is little talk about its limitations or possible hurdles. Recognizing these factors would offer a more even-handed evaluation of the approach The language can be a bit wordy and repetitive in spots. Making the wording simpler and keeping the terminology consistent would make it easier to read and understand. The conclusion could be improved by summarizing the main findings and the impact of the study more directly. It would also be helpful to suggest future research directions to give the work a sense of closure and context.
The author’s answer:
The linguistic presentation of the entire text was checked and revised, and the presentation of the conclusions was revised in the revised version of the manuscript. The shortcomings in this field, and next research direction have been elaborated.
We believe that the changes we have made address the concerns raised and improve the quality of the manuscript. Please review the revised version and all major changes have been highlighted in the manuscript. We are grateful for your thorough review and look forward to your feedback.
Thank you once again for your time and expertise.
Yours sincerely

Round 2
Reviewer 1 Report
Comments and Suggestions for Authors
accept
Reviewer 2 Report
Comments and Suggestions for Authors
The revised version is suitable for publication.